# Redox Chemistry of Pt(II) Complex with Non-Innocent NHC Bis(Phenolate) Pincer Ligand: Electrochemical, Spectroscopic, and Computational Aspects

Ilya K. Mikhailov [1,2], Zufar N. Gafurov [1,*], Alexey A. Kagilev [1,2], Vladimir I. Morozov [1], Artyom O. Kantyukov [1,2], Ekaterina M. Zueva [1,3], Gumar R. Ganeev [2], Ilyas F. Sakhapov [1], Asiya V. Toropchina [1], Igor A. Litvinov [1], Galina A. Gurina [4], Alexander A. Trifonov [4,5], Oleg G. Sinyashin [1] and Dmitry G. Yakhvarov [1,2,*]

[1] Arbuzov Institute of Organic and Physical Chemistry, FRC Kazan Scientific Center, Russian Academy of Sciences, Arbuzov Street 8, Kazan 420088, Russia; tiimhailovilya@gmail.com (I.K.M.); al-kagilev@mail.ru (A.A.K.); mmoorroozz2004@mail.ru (V.I.M.); kant.art@mail.ru (A.O.K.); zueva_ekaterina@mail.ru (E.M.Z.); sakhapovilyas@mail.ru (I.F.S.); toropchina@iopc.ru (A.V.T.); litvinov@iopc.ru (I.A.L.); oleg@iopc.ru (O.G.S.)
[2] Alexander Butlerov Institute of Chemistry, Kazan Federal University, Kremlovskaya Street 18, Kazan 420008, Russia; farsiov.bope@gmail.com
[3] Department of Inorganic Chemistry, Kazan National Research Technological University, Karl Marx Street 68, Kazan 420015, Russia
[4] Institute of Organometallic Chemistry of Russian Academy of Sciences, Tropinina Street 49, GSP-445, Nizhny Novgorod 603950, Russia; live_love_peace@mail.ru (G.A.G.); trif@iomc.ras.ru (A.A.T.)
[5] A.N. Nesmeyanov Institute of Organoelement Compounds, Russian Academy of Sciences (INEOS RAS), Vavilova Street 28, Moscow 119991, Russia
* Correspondence: gafurov.zufar@iopc.ru (Z.N.G.); yakhvar@iopc.ru (D.G.Y.)

**Abstract:** A Pt(II) complex bearing chelating tridentate bis-aryloxide tetrahydropyrimidinium-based N-heterocyclic carbene (NHC) was synthesized and characterized by using different techniques. Both cyclic voltammetry and differential pulse voltammetry were used to study the electrochemical properties of the complex, revealing two reversible one-electron oxidation processes. The chemical generation and isolation of one-electron-oxidized species were performed oxidizing the initial complex by means of AgBF$_4$. A combination of spectroscopic (UV-Vis/NIR- and EPR-) and theoretical (density functional theory (DFT)) studies suggests the formation of a Pt(II)-phenoxyl radical complex. The latter open-shell derivative was structurally characterized by means of X-ray diffraction analysis. Finally, the neutral platinum complex was tested as a mediator in the process of electrocatalytic oxidation of 2-(methylamino)ethanol (MEA).

**Keywords:** cyclic voltammetry; electron paramagnetic resonance; DFT calculations; UV-VIS/NIR; phenoxyl radical; pincer complex; platinum complex; NHC ligand

## 1. Introduction

Transition-metal complexes with redox-active ligands have been extensively studied over the past few decades due to their unique properties and intriguing chemical behavior [1–8]. They are widely applied in various fields of coordination chemistry including catalysis, organic synthesis, and material science [1,9–18]. Recent attempts in controlling catalytic processes employing redox non-innocent ligands are mostly inspired by the metal-radical motifs in active sites of many metalloenzymes [5], such as galactose oxidase, which contains one copper ion and converts a primary alcohol to the aldehyde in the presence of dioxygen [19,20]. The formation of a Cu(II)-phenoxyl radical species was proposed to be the key step in this transformation [21].

Indeed, electron-rich sterically hindered phenolates are among the most common non-innocent ligands that can assist in catalytic transformations by storing (in form of phenoxyl)

and delivering charge during catalytic transformations [22–27]. However, bidentate ligand scaffolds utilized in many catalytic reactions involving such phenolate ligands (or their aminophenol derivatives) often lead to unfavorable isomerization processes of the ligand field, precluding further transformations of a substrate [28,29]. In order to overcome the limitation the redox-active phenolates were merged with N-heterocyclic carbenes (NHCs), forming a pincer-type tridentate ligand [30,31]. NHCs are well-known as an important family of ligands in coordination chemistry and homogeneous catalysis due to their unique steric and electronic properties. Moreover, the strongly σ-donating nature makes the ligand more stable in its oxidized forms, while the strong *trans* effect provide hemilability of the auxiliary ligand, generating a vacant coordination site at the metal center which is a prerequisite for efficient substrate coordination and transformation under the homogeneous catalysis conditions [32–34]. In this context, the previously obtained diphenolate imidazolyl and benzimidazolyl carbenes were applied for constructing transition metal complexes with the coordinated pro-radical ligand [35–38]. The oxidation of the above-mentioned species allowed the formation of an electronic structure in which an unpaired electron is more likely localized on the ligand. However, the precise determination of the metal oxidation state in transition-metal complexes is not always possible. In reality, metal ions and ligands are both affected by the oxidation and the resulting spin density is shared over both, allowing a multi-configurational state [9,25]. Therefore, a careful examination of metal complexes via various spectroscopic methods combined with computational studies is required for a better description of their electronic structure, which is essential for the design of catalytic cycles with participation of redox non-innocent ligands [39,40].

Our research team has been focusing on the design, synthesis, and catalytic application of pincer-type ligands and their complexes of group 10 metals during the past few years [41–47]. The intrinsic diamagnetism of this metal ions in $d^8$ configuration allows for fine assignment of the electronic structure of the oxidized species. At the same time, the natural content of spin-1/2 isotope for platinum ($^{195}$Pt, 33.775%) gives the opportunity for careful examination of the electronic structure by EPR spectroscopy. Thus, herein we report on the synthesis of Pt(II) complex bearing tridentate diphenolate NHC ligand. The one-electron-oxidized species are generated chemically and electrochemically. Characterization of their electronic structures using combined UV-Vis/NIR- and EPR-spectroscopy; X-ray diffraction and DFT studies suggests the redox non-innocence of the ligand. The obtained results allowed the design of an electrocatalytic process for 2-(methylamino)ethanol oxidation with participation of Pt(II)-phenoxyl radical complex.

## 2. Results and Discussion

The diphenolate NHC precursor **L$^{H3}$Cl** was prepared according to our previous report [48]. The synthesis of its platinum complex was performed using modified procedure suggested by Mauro, Dagorne, Bellemin-Laponnaz, and co-authors for the related diphenolate imidazolyl and benzimidazolyl carbene complexes [30]. However, it should be noted, that in our case providing the synthesis at 100 °C (as reported by the authors) led to unfavorable deposition of black precipitate (more likely metallic platinum). Therefore, the tetrahydropyrimidin-1-ium based pro-ligand was treated with Pt(COD)Cl$_2$ and an excess of a base in pyridine at 50 °C for 20 h (Scheme 1). The isolation of the complex via filtration of the CH$_2$Cl$_2$ solution of crude material through a silica gel plug with further evaporation of the solvent afforded the platinum complex **Pt(L)Py** in 81% yield as an orange solid. The formation of the corresponding diphenolate NHC complex was confirmed by the presence of a carbene signal at δ$_C$ 159.2 ppm in the $^{13}$C{$^1$H} NMR spectrum, while no residual tetrahydropyrimidinium and phenol moieties were detected via $^1$H NMR spectroscopy analysis. The proposed formulation was also confirmed by elemental analysis and MALDI mass spectrometry.

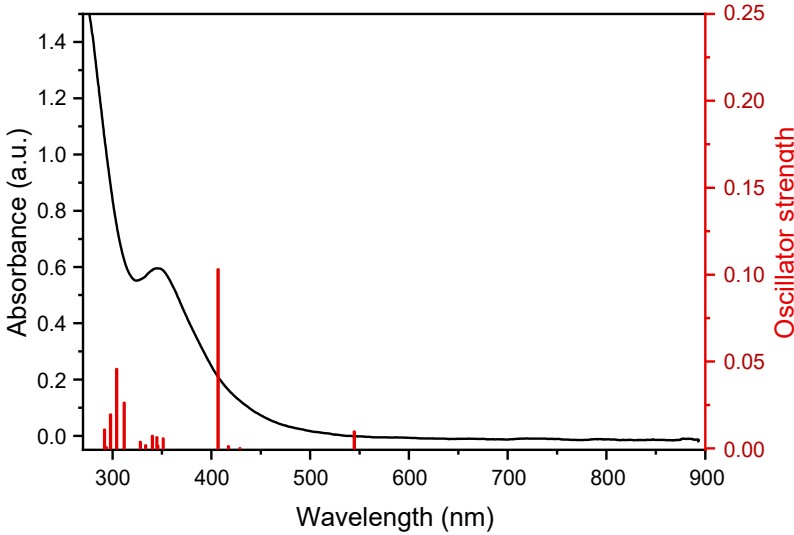

**Scheme 1.** Synthesis of Platinum Complex **Pt(L)Py**.

The UV-Vis spectrum of **Pt(L)Py** in $CH_2Cl_2$ (Figure 1) shows a weak absorption band at ca. 350 nm ($\varepsilon$ = 6000 $M^{-1} \cdot cm^{-1}$). According to time-dependent DFT calculations (Figure S4 and Table S1), the observed absorption originates from the electronic transition composed of a phenolate-to-NHC intraligand excitation mixed with a metal-to-metal transition (HOMO $\rightarrow$ LUMO + 2). It should also be noted that no NIR transitions were observed for **Pt(L)Py** complex (see Supplementary Materials for details).

**Figure 1.** UV-Vis spectrum of 0.1 mM solution of **Pt(L)Py** in $CH_2Cl_2$ at 298 K. The vertical bars represent the calculated electronic excitations.

The electrochemical properties of **Pt(L)Py** have been studied using the methods of cyclic voltammetry (CV) and differential pulse voltammetry (DPV) in $CH_2Cl_2$ in the presence of 0.1 M tetra-*n*-butylammonium tetrafluoroborate as supporting electrolyte. CV curve of complex **Pt(L)Py** exhibits two reversible oxidation peaks at $E_{1/2}^1$ = 0.25 V and $E_{1/2}^2$ = 0.80 V vs. the ferrocenium/ferrocene external standard (Figure 2). Analysis of the DPV curve's morphology provided evidence of the number of electrons involved. Thus, the values of 1.060 and 1.004 were obtained for the first and second oxidation, respectively. It may be concluded that the two oxidation waves occurred as a result of ligand-based oxidation processes that produced mono- and possibly bis-phenoxyl radical species [49,50]. No reduction peak observed when scanning towards the cathodic values of potentials in $CH_2Cl_2$. It is interesting to note that the reversibility of the second oxidation peak disappears when polar coordinating solvent is used (DMF, $CH_3CN$). At the same time, DMF enabled determination of the electroreduction process for the complex at −2.80 V, which was ascribed to the $Pt^{II} \rightarrow Pt^0$ conversion (see Supplementary Materials for details).

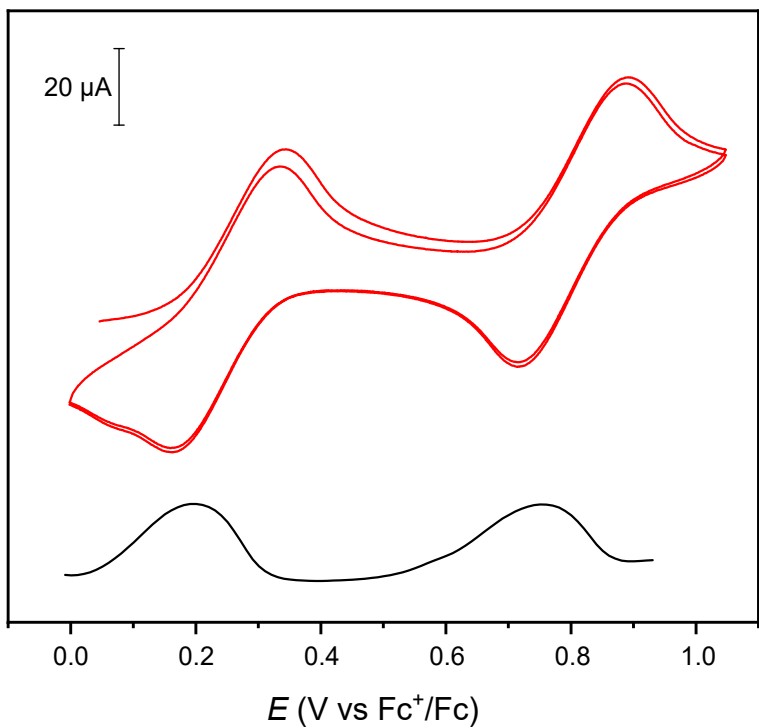

**Figure 2.** CV curve (red line) and DPV curve (black line) of 0.5 mM solution of the **Pt(L)Py** complex in $CH_2Cl_2$ with *n*-$Bu_4NBF_4$ (0.1 M) at a glass carbon electrode. Scan rate, 100 mV/s; reference, $Fc^+$/Fc; T = 298 K. The potential scanning from 0.00 to +1.00 V, back to 0.00 V, second cycle further to +1.00 V, and back to 0.00 V. The current scale is indicated for the CV curve.

The relatively large separation between the halfwave potentials for both processes ($\Delta E_{1/2}$ = 0.55 V) suggests that the first oxidation of a phenolate fragment in the complex significantly affects the second one. Thus, in order to evaluate the nature of the species formed in the electrooxidation process, the "chemical" oxidation of **Pt(L)Py** with $AgBF_4$ as an oxidizing agent ($E_{1/2}$ = 0.65 V vs. $Fc^+$/Fc [51]) in dry $CH_2Cl_2$ at room temperature under an inert atmosphere of nitrogen was performed (Scheme 2). The solution's change in color from orange to dark green and the precipitation of $Ag^0$ at the flask's bottom verified the successful oxidation of **Pt(L)Py**. The product **[Pt(L)Py][BF₄]** was isolated in 63% yield as an air-stable dark green solid. The proposed composition was confirmed via MALDI mass spectrometry using the characteristic ions with *m/z* 792.4 (positive mode) and *m/z* 87.1 (negative mode) corresponding to the **[Pt(L)Py]⁺** and [BF₄]⁻ species, respectively. The elemental analysis is also in agreement with the expected product.

**Scheme 2.** Synthesis of Platinum Complex **[Pt(L)Py][BF₄]**.

X-ray-quality dark green needle-like crystals were obtained via slow diffusion of pentane into a saturated solution of the formed product in dichloromethane at ca. 0 °C, and the structure of **[Pt(L)Py][BF₄]** was determined by means of X-ray diffraction. The complex

crystallizes as $CH_2Cl_2$ adduct in the $P\bar{1}$ (2) triclinic space group containing two molecules per unit cell. As depicted in Figure 3, the coordination geometry around the metal center is distorted-square-planar. Thus, the pyridine ligand occupies the coordination site *trans* to the pincer carbene carbon, while two oxygen atoms complete the metal coordination sphere standing *trans* to each other. The Pt−$C_{carbene}$ distance (1.954(2) Å) falls in the same range as those observed for related structures [30,52–56]. A significant distortion of the planarity of the {OCO}Pt moiety of the complex is observed (O(1)−N(1)−N(2)−O(2) = −14.63(8)°). The Pt−N bond length (2.094(2) Å) lines up with a *trans* influence of the carbene ligand (see Ref. [57] for example of **[Pt(Py)₄]²⁺** complex, where the average Pt–N bond length is 2.024 Å). Selected bond distances and angles are listed in the figure description.

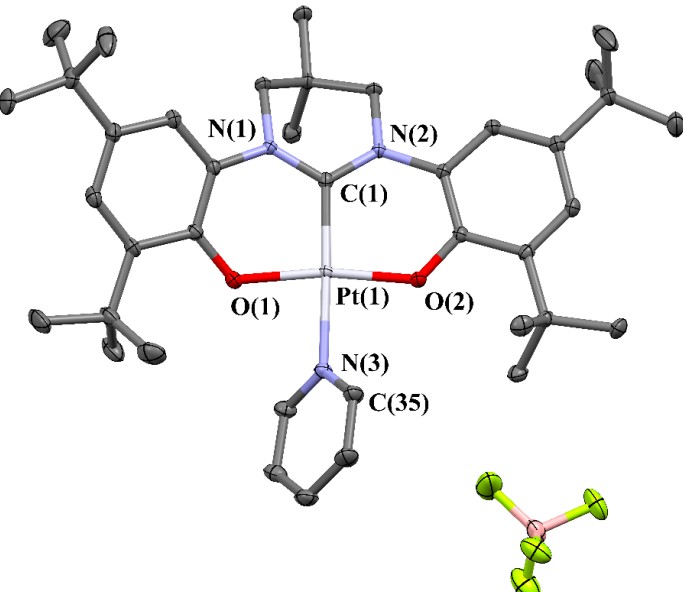

**Figure 3.** Molecular structure of the complex **[Pt(L)Py][BF₄]** (ellipsoids at 50% probability). Hydrogen atoms and $CH_2Cl_2$ molecules are omitted for clarity. Selected bond distances (Å) and angles (deg): C(1)−Pt(1), 1.954(2); O(1)−Pt(1), 1.937(1); O(2)−Pt(1), 1.923(1); N(3)−Pt(1), 2.094(2); C(1)−Pt(1)−O(1), 95.03(7); C(1)−Pt(1)−O(2), 94.57(7); O(1)−Pt(1)−O(2), 170.38(6); C(1)−Pt(1)−N(3), 177.64(7); O(1)−Pt(1)−N(3), 83.21(6); O(2)−Pt(1)−N(3), 87.22(6); and C(35)−N(3)−Pt(1)−O(2), 57.6(2).

An EPR analysis of a solution of **[Pt(L)Py][BF₄]** in $CH_2Cl_2$ at 298 K shows a signal at g = 2.070 with signal coupling coming from the $^{195}$Pt having nuclear spin of $^1/_2$ and natural content of 33.775%. (Figure 4a). The g value is considerably greater than what is commonly predicted for free phenoxyl radicals (typically 2.005) [58], although it is lower than the typical average g values reported for paramagnetic platinum complexes. Variable-temperature EPR study (Figure 4b) revealed the g-anisotropy. Thus, the frozen solution EPR spectrum of **[Pt(L)Py][BF₄]** in $CH_2Cl_2$ exhibits a highly anisotropic S = $\frac{1}{2}$ signal at $g_1$ = 1.943, $g_2$ = 2.123, and $g_3 \approx$ 2.149 ($g_{iso} \approx$ 2.071), that together with large g value suggests a considerable contribution of the metal d orbital to the SOMO [12,59]. Taken together, these observations provide strong evidence that the distribution of the unpaired electron in **[Pt(L)Py][BF₄]** is not uniform, and allow us to claim that the radical is localized over the ligand.

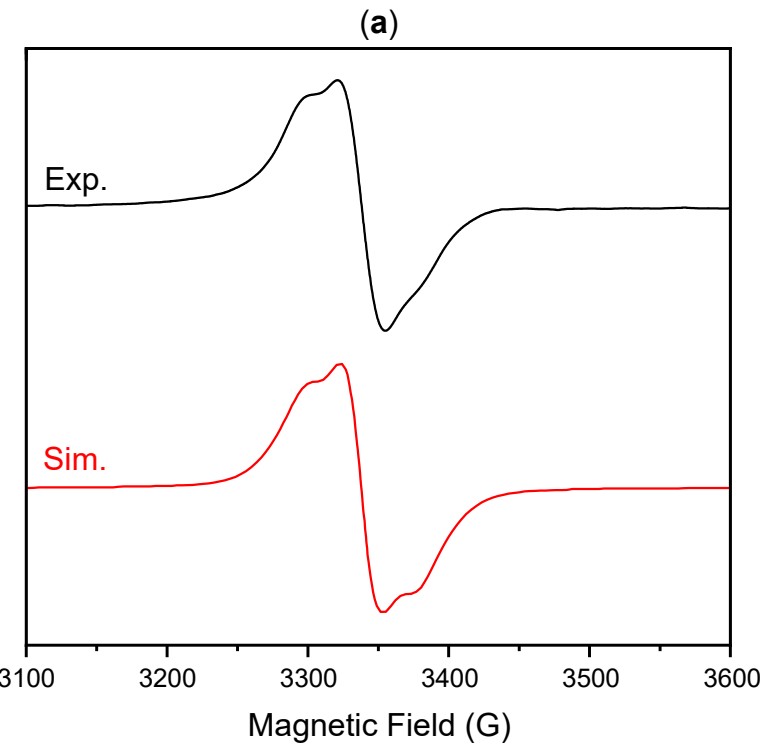

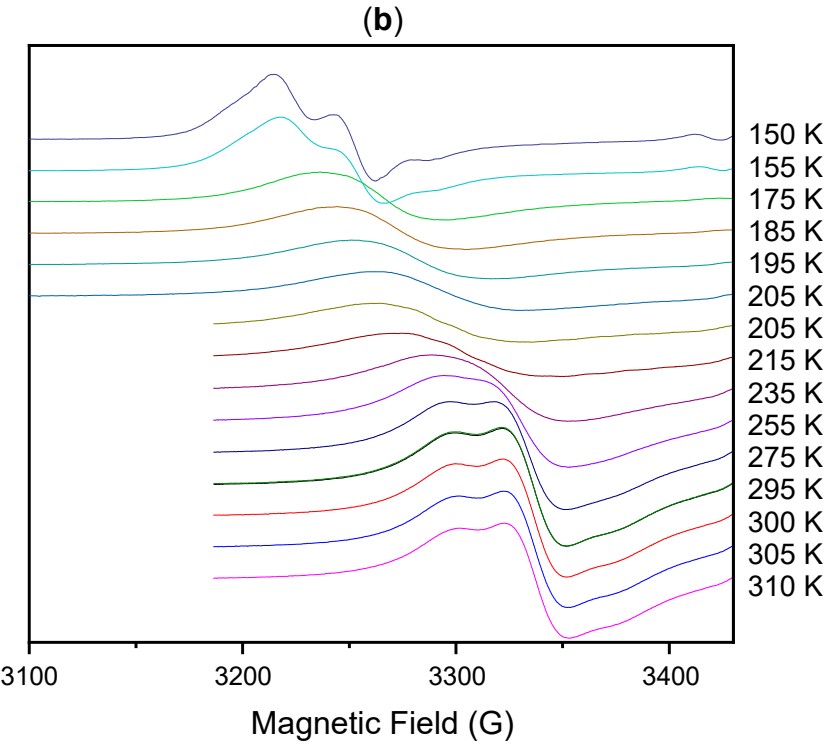

**Figure 4.** (**a**) X-Band EPR spectrum of 0.5 mM solution of **[Pt(L)Py][BF₄]** in $CH_2Cl_2$. Black line: experimental spectrum; red line: simulation by considering g value of 2.070, a Lorentzian/Line width of 60.464/22.620 G, $A_{Pt}$ = 64.865 G (considering 31.332% rel. conc. of [195]Pt); T = 298 K. (**b**) Variable-temperature EPR spectra of 0.5 mM solution of **[Pt(L)Py][BF₄]** in $CH_2Cl_2$.

The Mulliken spin–density distribution in **[Pt(L)Py]⁺** calculated at the DFT-optimized structure (Figure 5, left) exhibits a predominant phenoxyl-radical character. Although the spin density is shared between the metal center and the ligand, its major contribution

is delocalized within the ligand. It should also be noted that the nitrogen atoms of the carbene linker hold a small spin population, in contrast to the diphenolate imidazolyl based nickel analogue, which does not feature any significant spin density [35]. Note that the computed spin–density plot is consistent with the average local ionization energy (ALIE) isosurface map computed for **Pt(L)Py** (Figure 5, right)**,** where the blue-colored regions have a relatively low ALIE value and are, therefore, the most vulnerable to electrophilic attack.

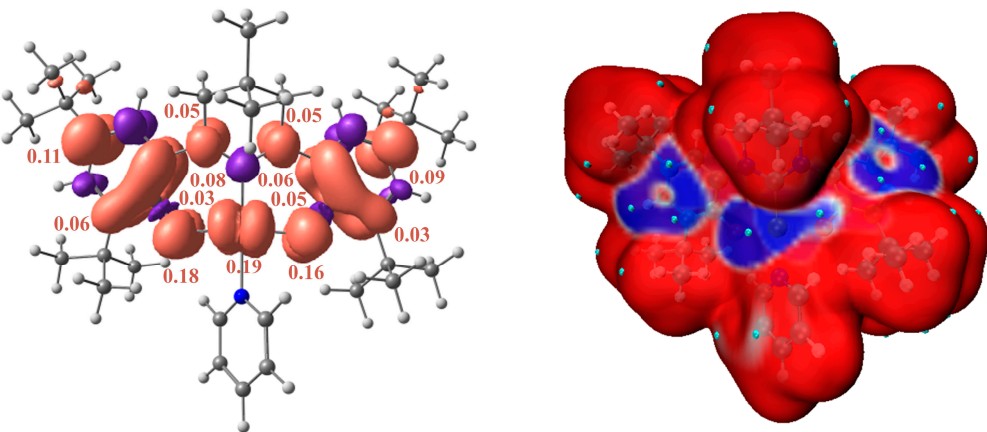

**Figure 5.** Spin–density plot computed for **[Pt(L)Py]$^+$** (**left**, isovalue = 0.001) and average local ionization energy isosurface map computed for **Pt(L)Py** (**right**, isovalue = 0.0005, blue dots represent the surface minima).

The UV-Vis and NIR spectra of **[Pt(L)Py][BF$_4$]** in CH$_2$Cl$_2$ are shown in Figure 6. In visible regions the complex exhibits an absorption band at ca. 466 nm, which is red-shifted compared to its neutral analogue. In addition, **[Pt(L)Py][BF$_4$]** shows a remarkable intense NIR band at 1242 nm ($\varepsilon$ = 7500 M$^{-1}$·cm$^{-1}$) with a narrow bandwidth at half height $\Delta\nu_{1/2}$ of 1712 cm$^{-1}$ (see Supplementary Materials for details).

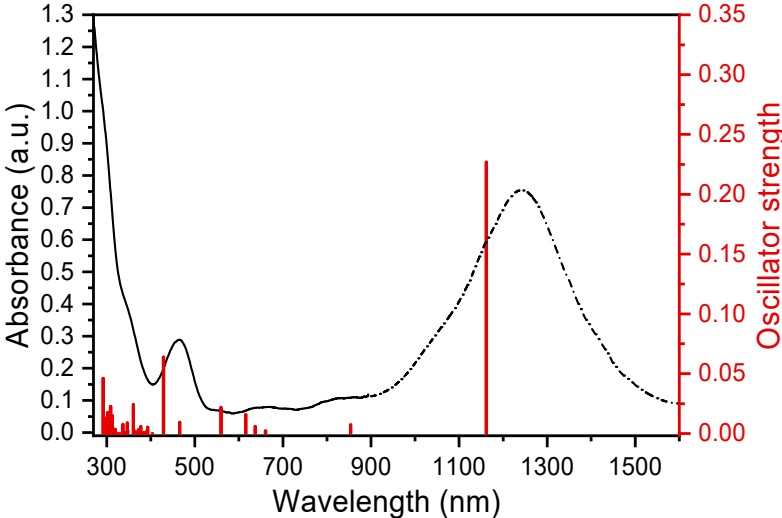

**Figure 6.** UV-Vis (solid black line) and NIR (dashed black line) spectra of 0.1 mM solution of **[Pt(L)Py][BF$_4$]** in CH$_2$Cl$_2$ at 298 K. The vertical red bars represent the calculated electronic excitations.

According to the Marcus–Hush relationship:

$$\Delta\nu_{HTL} = \sqrt{16\ln 2\ RT\ \nu_{max}} \tag{1}$$

Using the $\nu_{max}$ = 8050 cm$^{-1}$ (1242 nm) the bandwidth calculated in the high temperature limit (HTL) is 4308 cm$^{-1}$, which differs significantly from the experimental value.

This observation is in line with Marcus–Hush theory's failure to predict the IVCT band form at the class II/III borderline [59]. Moreover, a sharp and intense NIR transition, where $\Delta\nu_{1/2} \leq 2000$ cm$^{-1}$, $\varepsilon \geq 5000$ M$^{-1}$ cm$^{-1}$, is an indicator that the electron hole is at least partly delocalized over the ligand scaffold [60]. Based on the high intensity and narrow bandwidth, the lowest energy band can be assigned to the intraligand HOMO (donor) to SOMO (radical) electronic transition. In order to validate this hypothesis, absorption properties of **[Pt(L)Py]$^+$** were studied using time-dependent DFT (Figure S4 and Table S1). The calculated energy of the NIR band with a large oscillator strength is 1162 nm ($f_{osc}$ = 0.23), which matches well with the experimental data (1242 nm). The principal excitation is indeed the $\beta$-HOMO $\rightarrow$ $\beta$-LUMO transition (97%), which is predominately $\pi$-$\pi^*$ intraligand in nature. The calculated electronic transition of a higher energy ($\lambda_{calc}$ = 429 nm) corresponds to the experimental band observed at 466 nm. The principal excitations that contribute to the band are $\alpha$-HOMO $\rightarrow$ $\alpha$-LUMO (58%) and $\alpha$-HOMO $\rightarrow$ $\alpha$-LUMO + 1 (32%) transitions.

Having all these results in hand, we decided to test the platinum complex **Pt(L)Py** as a mediator in the process of electrocatalytic oxidation of amines. It is well known that the electrooxidation of amines is an important alternative to current chemical approaches, providing critical pathways for the synthesis and modification of a wide range of chemically valuable compounds, including pharmaceuticals and agrochemicals [61]. Details on the mechanistic aspects of the electrochemical oxidation of aliphatic amines can be found elsewhere [61]. However, the direct electrooxidation often leads to a passivation of the working electrode surface by the formation of a polymer film, which can sharply decrease current efficiency [62]. Indirect electrooxidation using an electron transfer mediator is an effective technique to avoid this problem [63]. Besides the minimization of the electrode surface fouling effect, the application of redox mediators allows a decrease in the overpotential needed for the direct oxidation of amines [61]. **Pt(L)Py**, which is prone to electrochemical oxidation to **[Pt(L)Py]$^+$**, meets the requirements usually addressed to a redox mediator (or catalyst): it has a reversible redox couple, while both the oxidized and reduced forms are relatively stable in the solution. Thus, 2-(methylamino)ethanol (MEA) was chosen as a model substrate for the catalytic tests. Figure 7 shows the CV curves, obtained for 1 mM solution of **Pt(L)Py** with different concentrations of MEA (0.00 M, 0.12 M and 0.24 M) in the range of 0.00 to 0.45 V vs. Fc$^+$/Fc in CH$_2$Cl$_2$ in the presence of $n$-Bu$_4$NBF$_4$ (0.1 M). It should be noted that MEA is electrochemically inactive in this region (see black curve in Figure 7). Reaching a concentration of MEA up to 0.12 M (see blue curve in Figure 7) leads to a disappearance of reversibility of the **[Pt(L)Py]$^+$**/**Pt(L)Py** couple with simultaneous increase in oxidation peak current. The peak oxidation current continues to increase until the substrate concentration reaches 0.24 M. At MEA concentrations greater than 0.24 M, saturation behavior is observed as the current becomes independent of the concentration of MEA. The efficiency of electrocatalytic MEA oxidation can be estimated via the ratio of the maximum catalytic current ($i_{cat}$) to the peak current ($i_p$) in the presence and absence of amine, respectively. Thus, the $i_{cat}/i_p$ value of 1.9 was obtained for **Pt(L)Py** as a mediator of this process.

These results are indicative of the electrochemical catalysis of secondary amine (MEA) oxidation via the active Pt(II)-phenoxyl radical complex oxidant. Further experiments related to reactivity and catalytic activity of platinum complexes **[Pt(L)Py]$^+$** and **Pt(L)Py** as well as other group 10 metals with diphenolate NHC ligand **L** are currently under progress.

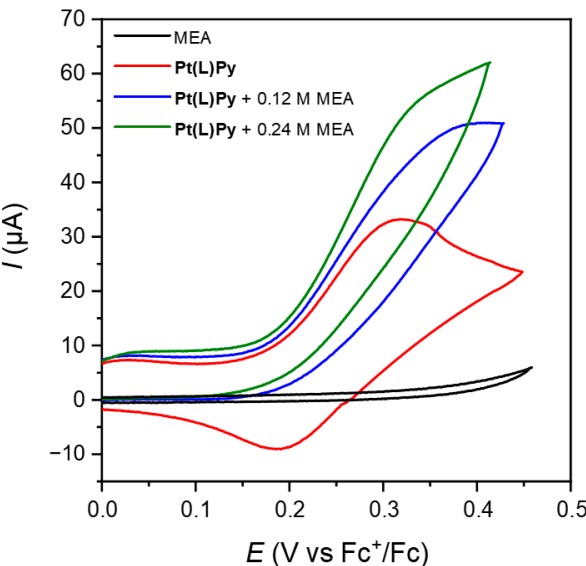

**Figure 7.** CV curves of 1 mM solution of the **Pt(L)Py** complex in $CH_2Cl_2$ with *n*-Bu₄NBF₄ (0.1 M) at a glass carbon electrode in the absence of MEA (red curve), in the presence of 0.12 M MEA (blue curve), in the presence of 0.24 M MEA (green curve). Black curve represents the CV of MEA in the absence of **Pt(L)Py**. Scan rate, 100 mV/s; reference, $Fc^+/Fc$; T = 298 K.

## 3. Materials and Methods

### 3.1. General Considerations

Standard Schlenk techniques have been applied for all reactions, which were carried out in a dry nitrogen environment. Organic solvents (pentane, $CH_2Cl_2$, pyridine, DMF, $CH_3CN$, and THF) were purified and degassed using standard procedures. $CDCl_3$ was degassed via freeze–pump–thaw cycles (three times) and kept over 3 Å molecular sieves before use. Ligand **L^{H3}Cl** was obtained using the previously described procedure [48]. All other reagents (AgBF₄ (Aldrich, St. Louis, MO, USA, 98%), *n*-Bu₄NBF₄ (Acros Organics, Geel, Belgium, 98%), ferrocene (Alfa Aesar, Karlsruhe, Germany, 99%), and Pt(COD)Cl₂ (Sigma-Aldrich, St. Louis, MO, USA, 99%)) were used without further purification.

NMR spectra were recorded at frequencies of 400.17 MHz ($^1H$) and 100.62 MHz ($^{13}C$) using a high-resolution BRUKER AVANCE-400 (Karlsruhe, Germany) spectrometer. $^1H$ and $^{13}C\{^1H\}$ chemical shifts are given downfield of tetramethylsilane in parts per million (ppm) and were calibrated against the resonance of the remaining protons of the used deuterated solvent. UV–Vis spectra were recorded on a SPECORD 50 PLUS Analytik Jena (Jena, Germany) spectrophotometer in 10 mm closed quartz cuvette at 298 K. NIR absorption spectra were recorded on a Bruker Vertex 70 spectrometer (Ettlingen, Germany) in 10 mm closed quartz cuvette at 298 K. The measurements related to the electron paramagnetic resonance (EPR) were carried out using X-band microwaves and a 100 kHz field modulator using a Bruker Elexsys E-500 spectrometer (Karlsruhe, Germany). MALDI-TOF studies were conducted using an Ultraflex III TOF/TOF (Bruker Daltonics, Karlsruhe, Germany) mass spectrometer equipped with a Nd:YAG laser. The spectra were measured in both positive and negative ion linear modes. *para*-Nitroaniline (PNA) was used as a matrix. The FlexControl software (Bruker Daltonik GmbH, Version 3.0) was used for instrument control and data acquiring. FlexAnalysis software (Bruker Daltonik GmbH, Version 3.0) was used to process the data. A high-temperature Elementar vario MACRO cube (Langenselbold, Germany) analyzer was used for the elemental analysis.

### 3.2. X-ray Structure Determination

The crystal structure of **[Pt(L)Py][BF₄]** was analyzed via X-ray diffraction utilizing a Bruker D8 QUEST (Karlsruhe, Germany) automated three-circle diffractometer. The diffractometer was equipped with a PHOTON III two-dimensional detector and an IμS

DIAMOND microfocus X-ray tube ($\lambda$ [Mo K$\alpha$] = 0.71073 Å) and operated under cooled conditions. The acquired diffraction data were processed using the APEX3 software program. The SHELXT software was used to solve the structure via the direct method [64] then the SHELXT program was used to improve it using the full-matrix least squares approach over $F^2$ [65]. WinGX software package [66] carried out all calculations. The geometry of the molecule and intermolecular interactions in the crystal were calculated using the PLATON program [67]. Molecular drawings were created with the ORTEP3 [66] and MERCURY [68] program. The non-hydrogen atoms were refined using the anisotropic approximation. The hydrogen atoms were placed in geometrically calculated positions and included in the refinement in the "riding" model. One of the solvate molecules of methylene chloride is disordered by the center of symmetry over two positions. The structure's crystallographic data were deposited at the Cambridge Crystallographic Data Center, and Table 1 lists the registration number and the key characteristics.

**Table 1.** Crystal data and structure refinement for complex **[Pt(L)Py][BF$_4$]**.

| **Moiety Formula** **Sum Formula** | **2(C$_{39}$H$_{55}$N$_3$O$_2$Pt), 2(BF$_4$), 3(CH$_2$Cl$_2$)** **C$_{81}$H$_{116}$B$_2$Cl$_{16}$F$_8$N$_6$O$_4$Pt$_2$** |
|---|---|
| formula weight | 2014.28 |
| temperature [K] | 100(2) |
| wavelength [Å] | 0.71073 |
| crystal system, space group | triclinic, $P\bar{1}$ *(No. 2)* |
| $a$ [Å] | 9.7661(12) |
| $b$ [Å] | 13.8600(15) |
| $c$ [Å] | 16.8963(19) |
| $\alpha$ [deg] | 102.107(3) |
| $\beta$ [deg] | 91.972(4) |
| $\gamma$ [deg] | 99.446(3) |
| $V$ [Å$^3$] | 2200.2(4) |
| $Z$, Dc [g cm$^{-3}$] | 1, 1.520 |
| absorption coefficient [mm$^{-1}$] | 3.424 |
| $F(000)$ | 1016 |
| crystal size [mm] | $0.30 \times 0.10 \times 0.02$ |
| $\Theta$ range for data collection [deg] | 2.4–32.0 |
| limiting indices | $-14 \leq h \leq 14, -20 \leq k \leq 20, -25 \leq l \leq 25$ |
| reflections measured | 113,538 |
| reflections unique | 15,266 |
| observed reflections [$I > 2\sigma(I)$] | 14,441 |
| GOF on F$^2$ | 1.093 |
| data/restraints/parameters | 15266/2/516 |
| final $R$ indices [$I > 2\sigma(I)$] | $R_1 = 0.0241$, w$R_2 = 0.0588$ |
| $R$ indices (all data) | $R_1 = 0.0262$, w$R_2 = 0.0593$ |
| largest diff. peak and hole [e Å$^{-3}$] | 2.02 and $-1.47$ |
| CCDC number | 2,277,791 |

### 3.3. Cyclic Voltammetry (CV) and Differential Pulse Voltammetry (DPV)

In CV and DPV experiments the concentration of the complex was 5·mM in CH$_2$Cl$_2$ as a solvent with (*n*-Bu$_4$N)BF$_4$ (0.1 M) as a supporting electrolyte. The electrochemical cell had three channels and included a glassy carbon (GC) working electrode with a surface area of 3.14 mm$^2$, a Pt wire auxiliary electrode with a diameter of 1 mm, and Ag/AgNO$_3$ (0.01 M solution in CH$_3$CN) as the reference electrode. The cell was maintained in an inert nitrogen environment, and all tests were performed with a working solution of 5 mL. The ferrocenium/ferrocene external standard was used to adjust the values. Prior to each experiment, the GC electrode was cleaned and polished using 0.05 μm aluminium oxide polishing paper. Curves were recorded with a E2P potentiostat from BASi Epsilon (West Lafayette, IN, USA) at a constant potential scan rate of 100 mV·s$^{-1}$. The equipment

comprises a measuring unit and a DellOptiplex 320 computer that operates the Epsilon-EC-USB-V200 software.

### 3.4. Experimental Procedures and Product Characterization

### 3.4.1. Synthesis of **Pt(L)Py**

A mixture of 1,3-bis(3,5-di-tert-butyl-2-hydroxyphenyl)-5,5-dimethyl-3,4,5,6-tetrahydropyrimidin-1-ium chloride $\mathbf{L^{H3}Cl}$ (100 mg, 0.179 mmol), Pt(COD)Cl$_2$ (85 mg, 0.227 mmol), and potassium carbonate (30 equiv, 744 mg, 5.4 mmol) was suspended in pyridine in a Schlenk flask. For 50 min, the resulting mixture was sonicated. The mixture was then stirred for 20 h at 50 °C in a nitrogen environment. The volatiles were removed from the resulting suspension under reduced pressure. The residue then dissolved in dichloromethane; the resulting solution was filtered through a Celite plug and concentrated under reduced pressure, affording complex **Pt(L)Py** (115 mg, 81%, orange solid). $^1$H NMR (CDCl$_3$, 400.17 MHz, 300 K): 1.11 (s, 18H, C(C$H_3$)$_3$), 1.29 (s, 18H, C(C$H_3$)$_3$), 1.31 (s, 6H, C(C$H_3$)$_2$), 3.73 (s, 4H, C$H_2$), 6.92 (d, $^4J_{HH}$ = 2.4 Hz, 2H, C$H_{Phenoxy}$), 7.01 (d, $^4J_{HH}$ = 2.4 Hz, 2H, C$H_{Phenoxy}$), 7.41 (td, $^3J_{HH}$ = 6.3, $^4J_{HH}$ = 1.4 Hz, 2H, C$H_{Py}$), 7.82 (tt, $^3J_{HH}$ = 7.7, $^4J_{HH}$ = 1.5 Hz, 1H, C$H_{Py}$), 8.81 (dd, $^3J_{HH}$ = 4.8, $^4J_{HH}$ = 1.5 Hz, 2H, NC$H_{Py}$). $^{13}$C{$^1$H}NMR (100.62 MHz, CDCl$_3$, 300 K) δ 24.72 (C(CH$_3$)$_2$), 28.40 (C(CH$_3$)$_2$), 29.76 (C(CH$_3$)$_3$), 31.69 (C(CH$_3$)$_3$), 34.30 (C(CH$_3$)$_3$), 35.29 (C(CH$_3$)$_3$), 58.41 (NCH$_2$), 115.68 (CH$_{Ar}$), 119.52 (CH$_{Ar}$), 124.65 (CH$_{Py}$), 136.15 (C–O), 138.01 (CH$_{Py}$), 138.03 (C$_{Ar}$), 138.16 (C$_{Ar}$), 151.67 (CH$_{Py}$ and C$_{Ar}$), and 159.37 (C$_{carbene}$). MALDI-TOF MS *m*/*z*: calcd for C$_{39}$H$_{55}$N$_3$O$_2$Pt [M]$^+$ 792.97, found 792.4. Anal. Calcd (%) for C$_{39}$H$_{55}$N$_3$O$_2$Pt·0.5CH$_2$Cl$_2$: C, 56.79; H, 6.76; N, 5.03. Found: C, 56.59; H, 6.78; N, 5.05.

### 3.4.2. Synthesis of [**Pt(L)Py**][**BF$_4$**]

In a Schlenk flask, **Pt(L)Py** (33 mg, 0.041 mmol) was dissolved in dichloromethane (7.5 mL). Afterwards, the solution of AgBF$_4$ (7.4 mg, 0.037 mmol) in dichloromethane (7.5 mL) was added dropwise to **Pt(L)Py** and the mixture was stirred for 1 h at room temperature. After the completion of the reaction, the solution was filtered, concentrated under reduced pressure, and washed three times with pentane (5 mL). Complex [**Pt(L)Py**][**BF$_4$**] was isolated in 63% yield (20.5 mg) as a dark green solid. Crystals suitable for X-ray diffraction analysis were grown by diffusion of pentane into a saturated solution of the product in dichloromethane at 0 °C. MALDI-TOF MS *m*/*z*: calcd for C$_{39}$H$_{55}$BF$_4$N$_3$O$_2$Pt [M]$^+$ 792.97, found 792.4. Anal. Calcd (%) for C$_{39}$H$_{55}$BF$_4$N$_3$O$_2$Pt·0.7CH$_2$Cl$_2$: C, 50.77; H, 6.05; N, 4.47. Found: C, 50.19; H, 7.00; N, 4.98.

### 3.5. Quantum-Chemical Calculations

Geometry optimizations were carried out using the PBE0/LANL2DZ computational procedure, which employs the PBE0 functional [69] in conjunction with the LANL2DZ basis set and associated ECP [70–73] for Pt atom (replaces its 60 core electrons). Absorption spectra were calculated via time-dependent DFT using the B3LYP functional [74,75] and a larger basis set (def2-TZVP [76–78]). All these calculations were done with the ORCA package (version 4.0) [79]. Average local ionization energy analysis was performed using the Multiwfn package [80].

## 4. Conclusions

In summary, we have reported the direct and good-yielding synthesis and characterization of a Pt(II) complex bearing tridentate diphenolate tetrahydropyrimidin-1-ium-based NHC ligand. The electrochemical behavior of the complex, investigated via voltammetric techniques, revealed the reversible redox wave at 0.25 V vs. Fc$^+$/Fc couple, which corresponds to the oxidation of the non-innocent phenolate moiety with the formation of Pt(II)-phenoxyl radical complex. The latter was obtained preparatively via "chemical" oxidation of neutral analogue by means of AgBF$_4$ and its electronic structure was determined using combined UV-Vis/NIR- and EPR-spectroscopy, X-ray diffraction, and DFT studies.

The obtained g value and spin–density plot suggest the predominant phenoxyl-radical character of the complex. The intense NIR band supports the delocalization of the unpaired electron over the ligand. It was found that neutral platinum complex is active in electro-catalytic oxidation of secondary amine (MEA) via the formation of active Pt(II)-phenoxyl radical complex oxidant with the $i_{cat}/i_p$ value of 1.9.

**Supplementary Materials:** The supporting information for this article can be downloaded at: https://www.mdpi.com/article/10.3390/catal13091291/s1, Figure S1: NIR spectrum of $CH_2Cl_2$ at 298 K, l = 1 cm; Figure S2: NIR spectrum of **Pt(L)Py** (c = 0.1 mM) in $CH_2Cl_2$ at 298 K, l = 1 cm; Figure S3: NIR spectrum of **[Pt(L)Py][BF_4]** (c = 0.1 mM) in $CH_2Cl_2$ at 298 K, l = 1 cm; Figure S4: The calculated absorption spectra for **Pt(L)Py** (top graph) and **[Pt(L)Py][BF_4]** (bottom graph). The vertical lines showing the position of electronic transitions and their intensity (*f*—oscillator strength) were broadened by the Lorentz function; Table S1: Selected TD-DFT-calculated excitation energies (absorption wavelengths), oscillator strengths, and main compositions of the most important electronic transitions for **Pt(L)Py** and **[Pt(L)Py][BF_4]**; Figure S5: CV curves recorded from a solution containing complex **Pt(L)Py** in different solvents (c = 5 mM) in the presence of (*n*-Bu$_4$N)BF$_4$ (0.1 M) at the scan rate of 100 mV·s$^{-1}$ on the GC working electrode (T = 298 K); Table S2: Peak potentials on the CV curves of complex **Pt(L)Py** in different solvents; Figure S6: $^1$H NMR (CDCl$_3$, 400.17 MHz, 300 K) spectrum of **Pt(L)Py**; Figure S7: $^{13}$C{$^1$H}NMR (100.62 MHz, CDCl$_3$, 300 K) spectrum of **Pt(L)Py**.

**Author Contributions:** Investigation, formal analysis, I.K.M., A.A.K., V.I.M., A.O.K., G.R.G., I.F.S., A.V.T., I.A.L. and G.A.G.; DFT calculations, E.M.Z.; writing—original draft preparation, Z.N.G.; writing—review and editing, I.K.M., G.A.G., E.M.Z. and D.G.Y.; conceptualization, methodology, supervision, and project administration, A.A.T., O.G.S. and D.G.Y. All authors have read and agreed to the published version of the manuscript.

**Funding:** This research was funded by the grant for support of the Leading Scientific Schools of the Russian Federation (Project No. 4078.2022.1.3), the Government assignment for FRC Kazan Scientific Center of RAS, and the Kazan Federal University Strategic Academic Leadership Program (PRIORITY-2030).

**Data Availability Statement:** All data are available from the authors upon request.

**Acknowledgments:** X-ray diffraction, elemental analysis, and spectrometric studies were carried out in the Distributed Spectral-Analytical Center of Shared Facilities for the Study of Structures, Compositions, and Properties of Substances and Materials, Kazan Scientific Center, Russian Academy of Sciences within the state task for the Kazan Scientific Center, Russian Academy of Sciences.

**Conflicts of Interest:** The authors declare no conflict of interest.

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
