# Peer review of "Redox Chemistry of Pt(II) Complex with Non-Innocent NHC Bis(Phenolate) Pincer Ligand: Electrochemical, Spectroscopic, and Computational Aspects"

_catalysts, doi:10.3390/catal13091291_

Round 1

Reviewer 1 Report

This is a very nice study of the synthesis and oxidation of a Pt(II)-OCO-pincer complex to a radical complex. The authors have very nicely shown the properties of this radical complex including and the distribution of the electron density of the radical.

Minor issue: The NMR spectra should be included into the Supporting Information. From the data provided in the experiemental section (l.321-324): two peaks are missing as well as an important Carbene-Pt coupling.

For both complexes, the H-value of the CHN analyses differs by 1.3 %. What is the reason? Please comment.

Provide the probability of the thermal distortion parameters in Figur 3.

The English is very good, however some minor errors were identified:

l.34: They "are widely" applied... , not "wildly" (maybe sometimes even that :-) ).

Check "lead" (l.83, l.216): if past tense write: "led", if presence check if it should be "leads"

l.48 ...tridentate "ligand".

l54, "under" the homogenous...

l.101, "It" should.

l262, recorded "on" a Bruker..

Author Response

Review #1:

Comments and Suggestions for Authors

This is a very nice study of the synthesis and oxidation of a Pt(II)-OCO-pincer complex to a radical complex. The authors have very nicely shown the properties of this radical complex including and the distribution of the electron density of the radical.

Minor issue: The NMR spectra should be included into the Supporting Information. From the data provided in the experiemental section (l.321-324): two peaks are missing as well as an important Carbene-Pt coupling.

Response: The NMR spectra were included into the Supporting Information. In fact, two carbons of pyridine ligand are difficult to distinguish because they overlap with the signals from OCO ligand. However, their chemical shifts were found and added to the paper. Unfortunately, no Carbene-Pt coupling was observed in the NMR spectrum. It should be noted that this situation is typical for this family of platinum complexes (see Organometallics 2014, 33, 4374−4384 for similar diphenolate imidazolyl and benzimidazolyl analogues).

For both complexes, the H-value of the CHN analyses differs by 1.3 %. What is the reason? Please comment.

Response: We repeated the elemental analysis for the purified complexes Pt(L)Py and [Pt(L)Py]BF4, but again we obtained deviations from the expected values. As observed for similar diphenolate imidazolyl and benzimidazolyl analogues (Organometallics 2014, 33, 4374−4384), we consider the formation of CH2Cl2 adducts of the complexes. Thus, for isolated powder samples of Pt(L)Py and [Pt(L)Py]BF4, the most suitable formulations are C39H55N3O2Pt∙0.5СH2Cl2 and C39H55BF4N3O2Pt∙0.7СH2Cl2, respectively (added to the paper).

Provide the probability of the thermal distortion parameters in Figur 3.

Response: The probability of the thermal distortion (50%) has been added to the description of Figure 3.

Comments on the Quality of English Language

The English is very good, however some minor errors were identified:

l.34: They "are widely" applied... , not "wildly" (maybe sometimes even that :-) ).

Check "lead" (l.83, l.216): if past tense write: "led", if presence check if it should be "leads"

l.48 ...tridentate "ligand".

l54, "under" the homogenous...

l.101, "It" should.

l262, recorded "on" a Bruker..

Response: We appreciate the thorough examination of our manuscript. The manuscript was carefully checked and all the errors have been corrected.

Reviewer 2 Report

Authors present the synthesis and charactirization of a chelating tridentate bis-aryloxide tetrahydropyrimidinium based N-heterocyclic carbene Pt(II) complex. This inorganic chemistry-lover referee thinks this is a very well executate piece of work which must be a reference of how inorganic chemistry must be carried out. Although the synthesis of the ligand and the procedure for the obtaning of the metal complex is not novel at all, the following characterization and the analisys of the results are impeccable.

My only concern/suggestion would be to add a preliminary results of the  catalytic studies due to this is the only thing I really missed. Additionally, It is a catalysis journal.

Author Response

Review #2:

Authors present the synthesis and charactirization of a chelating tridentate bis-aryloxide tetrahydropyrimidinium based N-heterocyclic carbene Pt(II) complex. This inorganic chemistry-lover referee thinks this is a very well executate piece of work which must be a reference of how inorganic chemistry must be carried out. Although the synthesis of the ligand and the procedure for the obtaning of the metal complex is not novel at all, the following characterization and the analisys of the results are impeccable.

My only concern/suggestion would be to add a preliminary results of the  catalytic studies due to this is the only thing I really missed. Additionally, It is a catalysis journal.

Response: We are thankful to the Reviewer for this remark. The platinum complex obtained was tested as a mediator in the process of electrocatalytic oxidation of 2-(methylamino)ethanol. The efficiency of electrocatalytic MEA oxidation was estimated via the ratio of the maximum catalytic current (icat) to the peak current (ip) in the presence and in the absence of amine, respectively. Thus, the icat/ip value of 1.9 was obtained for Pt(L)Py as the electrocatalyst for this process.

Reviewer 3 Report

The manuscript reported a combined experimental and computational study of the redox properties of a Pt(II) complex with pincer ligand NHC bis(phenolate). The computational section brought insights into the electronic properties of the Pt complex. However, there are some issues with the computational methods and data presentation. The questions should be carefully addressed before publication.

1.      Line 340. The exchange-correlation functional was switched from PBE0 to B3LYP. What is the purpose of changing functional? Both PBE0 and B3LYP are hybrid functionals with similar HF exchange ratios. Using B3LYP usually will not significantly improve the quality of the TDDFT calculations.

2.      I suggest adding the iso-value in the figures captions for all calculated iso-surfaces. Such as Figure 5, Table S1.

3.      The broadening energies for all calculated spectra should be specified in the captions.

4.      The authors have demonstrated abundant information on the optical absorption properties using TDDFT. However, the theme is redox chemistry. I suppose that there are more descriptors related to the redox properties. I do not want to bring extra burdens to the authors. But I still want to suggest some descriptors that can be scanned quickly using the current geometry and wavefunctions. For example, the average local ionization energy (ALIE) can be used to examine the energy for taking electrons out of the molecules, which can reveal the oxidation active sites on the molecule. A similar analysis, such as the local electron affinity (LEA), can be done to reveal electrophilic regions. Most of these analyses are collected and implemented in the Multiwfn package.

5.      Line 118. Is there any computational evidence for the Pt(II) to Pt(0) transition for example electron transfer?

The majority of the manuscript is fine. Only minor revisions are needed. 

Author Response

Review #3:

The manuscript reported a combined experimental and computational study of the redox properties of a Pt(II) complex with pincer ligand NHC bis(phenolate). The computational section brought insights into the electronic properties of the Pt complex. However, there are some issues with the computational methods and data presentation. The questions should be carefully addressed before publication.

1) Line 340. The exchange-correlation functional was switched from PBE0 to B3LYP. What is the purpose of changing functional? Both PBE0 and B3LYP are hybrid functionals with similar HF exchange ratios. Using B3LYP usually will not significantly improve the quality of the TDDFT calculations.

Response: In this manuscript, we used the computational protocol proposed earlier in series of papers dealing with the assessment of absorption and emission characteristics of various organometallic complexes (e.g., Dalton Trans., 2020, 49, 482–491; Dalton Trans., 2021, 50, 13421–13429; Inorg. Chem. 2021, 60, 9, 6804–6812; etc.). The PBE0/LANL2DZ computational procedure was found to be best-suited for geometry optimizations (the B3LYP functional provided a worse agreement with the X-ray structure). Energetic characteristics were estimated using the B3LYP functional and a larger basis set (def2-TZVP). The B3LYP functional is commonly used in TDDFT calculations, but we agree that the PBE0 functional would provide similar results.

2) I suggest adding the iso-value in the figures captions for all calculated iso-surfaces. Such as Figure 5, Table S1.

Response: Thank you. The required information has been added.

3) The broadening energies for all calculated spectra should be specified in the captions.

Response: Thank you. The required information has been added.

4) The authors have demonstrated abundant information on the optical absorption properties using TDDFT. However, the theme is redox chemistry. I suppose that there are more descriptors related to the redox properties. I do not want to bring extra burdens to the authors. But I still want to suggest some descriptors that can be scanned quickly using the current geometry and wavefunctions. For example, the average local ionization energy (ALIE) can be used to examine the energy for taking electrons out of the molecules, which can reveal the oxidation active sites on the molecule. A similar analysis, such as the local electron affinity (LEA), can be done to reveal electrophilic regions. Most of these analyses are collected and implemented in the Multiwfn package.

Response: Thank you very much for the proposed suggestions! The ALIE-analysis data are now included in the manuscript. Indeed, they reinforce the description of the [Pt(L)Py] → [Pt(L)Py]+ oxidation process discussed in the paper.

5) Line 118. Is there any computational evidence for the Pt(II) to Pt(0) transition for example electron transfer?

Response: Yes, test calculations (not included in the paper) revealed that all metal-based orbitals are occupied in the [Pt(L)Py]2- dianion (d10 configuration).

Round 2

Reviewer 1 Report

All necessary revisions have been carried out. I recommend acceptance of this manuscript.

Author Response

Response: Thank you!

Reviewer 3 Report

* Figure 5-Right. The author showed the ALIE results. It should be further specified that the blue dots represented the ALIE extrema.

* I am sure that the author clearly knows the electrons included in the LANL2 ECP. However, It would be helpful to explicitly point out the electrons included in the ECP.

The writing satisfied the requirement for publication.

Author Response

* Figure 5-Right. The author showed the ALIE results. It should be further specified that the blue dots represented the ALIE extrema.

Response: The required information has been added.

* I am sure that the author clearly knows the electrons included in the LANL2 ECP. However, It would be helpful to explicitly point out the electrons included in the ECP.

Response: The required information has been added.